# Measures Derived from Panoramic Ultrasonography and Animal-Based Protein Intake Are Related to Muscular Performance in Middle-Aged Adults

**DOI:** 10.3390/jcm10050988

**Published:** 2021-03-02

**Authors:** Nathaniel R. Johnson, Christopher J. Kotarsky, Kyle J. Hackney, Kara A. Trautman, Nathan D. Dicks, Wonwoo Byun, Jill F. Keith, Shannon L. David, Sherri N. Stastny

**Affiliations:** 1Department of Health, Nutrition, and Exercise Sciences, North Dakota State University, Fargo, ND 58105, USA; nathaniel.johnson.4@ndsu.edu (N.R.J.); Kyle.hackney@ndsu.edu (K.J.H.); shannon.david@ndsu.edu (S.L.D.); 2Department of Health and Human Physiological Sciences, Skidmore College, Saratoga Springs, NY 12866, USA; ckotarsk@skidmore.edu; 3Department of Health and Exercise Science, Gustavus Adolphus College, St. Peter, MN 56082, USA; karat@gustavus.edu; 4Department of Nutrition, Dietetics and Exercise Science, Concordia College, Moorhead, MN 56562, USA; ndicks@cord.edu; 5Department of Health and Kinesiology, University of Utah, Salt Lake City, UT 84112, USA; won.byun@utah.edu; 6Department of Family and Consumer Sciences, University of Wyoming, Laramie, WY 82071, USA; jkeith5@uwyo.edu

**Keywords:** panoramic ultrasound, echogenicity, specific force, isokinetic dynamometry, protein intake, muscle quality, strength, endurance

## Abstract

Ultrasonography advantageously measures skeletal muscle size and quality, but some muscles may be too large to capture with standardized brightness mode (B-mode) imaging. Panoramic ultrasonography can capture more complete images and may more accurately measure muscle size. We investigated measurements made using panoramic compared to B-mode ultrasonography images of the rectus femoris with muscular performance. Concurrently, protein intake plays an important role in preventing sarcopenia; therefore, we also sought to investigate the association between animal-based protein intake (ABPI) and muscular performance. Ninety-one middle-aged adults were recruited. Muscle cross-sectional area (CSA) and thickness were obtained using B-mode and panoramic ultrasound and analyzed with Image J software. Muscular performance was assessed using isokinetic dynamometry, a 30-s chair test, and handgrip strength. Three-day food diaries estimated dietary intakes. Linear regression models determined relationships between measures from ultrasonography and muscular performance. Mixed linear models were used to evaluate the association between ABPI and muscular performance. Muscle CSA from panoramic ultrasonography and ABPI were positively associated with lower-body strength (β ± S.E.; CSA, 42.622 ± 20.024, *p* = 0.005; ABPI, 65.874 ± 19.855, *p* = 0.001), lower-body endurance (β ± S.E.; CSA, 595 ± 200.221, *p* = 0.001; ABPI, 549.944 ± 232.478, *p* = 0.020), and handgrip strength (β ± S.E.; CSA, 6.966 ± 3.328, *p* = 0.004; ABPI, 0.349 ± 0.171, *p* = 0.045). Panoramic ultrasound shows promise as a method for assessing sarcopenia. ABPI is related to better muscular performance.

## 1. Introduction

Earlier and more frequent assessments of muscle strength, mass, size, and quality and physical performance could help prevent sarcopenia by indicating a need for treatment or other intervention. According to the European Working Group on Sarcopenia in Older People 2, low muscle strength is the first criteria of sarcopenia, and low muscle mass or quality is the second; both must be assessed to determine sarcopenia [1]. Low physical performance in addition to low muscle strength and quantity is considered severe sarcopenia [1]. Measures of muscle strength, such as handgrip strength and physical performance (e.g., 30-s chair stand), however, can be performed with minimal equipment and are used across various settings [1]. Although several methods can be used to accurately assess muscle quantity and quality such as computed tomography (CT), magnetic resonance imaging (MRI), and dual x-ray absorptiometry, these techniques require expensive equipment and are not portable, limiting their utility. Ultrasonography is a portable and relatively low-cost method of assessing muscle size [2], making it a potentially useful tool for evaluating sarcopenia for clinical or research purposes [3,4]. Beyond this, ultrasonography records a measure of muscle quality in the form of echogenicity or echo intensity [5,6,7], making ultrasound a potentially more powerful tool than bioelectrical impedance for assessing sarcopenia or signs of pre-sarcopenia in middle age.

Others have used ultrasonography to successfully diagnose sarcopenia [7,8,9]. However, two of these studies were performed with either frail elderly patients or older adults diagnosed with chronic kidney disease [8,9]. Not only are the causes of sarcopenia thought to start earlier in life [1], making middle-aged adults a population of interest, but also, older adults often have smaller muscles that can be captured using a traditional ultrasound image at 50% of leg length. Although Ismail and colleagues [7] were able to discriminate between those with sarcopenia and those without in a younger cohort, they did this by using longitudinal and not transverse images of the rectus femoris. The crux of the issue is that in populations that have greater muscle mass at the midpoint of the thigh, such as younger populations, the entire transverse rectus femoris may be too large to capture in one image [10]. Assuming the goal is to image the entire transverse rectus femoris, then there are two workarounds: one is to use a feature, like the panoramic feature, to record the entire rectus femoris at the midpoint of the thigh, and the other is to move the imaging site distally down the leg where the rectus femoris has smaller transverse sections. Other researchers have validated panoramic ultrasound of the quadriceps with MRI [11], but to our knowledge, the relationship between ultrasonographic measures of the transverse rectus femoris captured using the panoramic feature and muscular performance, in particular that of the knee extensors, has not been investigated. Because muscle strength is more closely related to sarcopenia than muscle mass [1,12], the association warranted investigation. 

Beyond this, specific force, the amount of force produced per unit of muscle, like echogenicity [6], is considered a measure of muscle quality [12]. Although echogenicity of the rectus femoris is related to muscle quality assessed using CT [8], and to a lesser extent knee extensor strength [13], the echogenicity of the rectus femoris has not been directly related to the specific force of the muscle. However, Ismail and colleagues [7] reported a significant relationship between echogenicity of rectus femoris and handgrip strength relative to bodyweight, a crude measure of specific force. If echogenicity and specific force reflect the muscle quality of the rectus femoris, then they should be closely related. We also sought to determine this relationship. 

Outside of assessing the condition, nutrition is another important consideration for preventing and treating sarcopenia. Although there are many nutritional factors that can impact sarcopenia [14], dietary protein is perhaps of greatest interest because of its ability to stimulate muscle protein synthesis [15]. Recently though, the role of protein intake in performance has come into question, with one group finding no relationship between protein intake and measures of muscular performance, such as handgrip strength, knee extensor strength, and 30-s chair stand test performance [16]. Foods from animal and plant sources, of course, differ in their digestibility and amino acid content [17], and therefore in their ability to stimulate muscle protein synthesis [18]. Due to the differential impact that animal-based protein has on muscle protein synthesis, we secondarily sought to determine the relationship between animal-based protein intake (ABPI) and lower-body strength and endurance, handgrip strength, and 30-s chair stand performance, measures of muscular performance.

## 2. Materials and Methods

This was a cross-sectional study conducted in the North Dakota State University Healthy Aging Lab from October 2016 to December 2018. A total of 50 women and 41 men from the local community were recruited using e-mail, flyers, and word-of-mouth to visit the research lab for two sessions. During the first session, anthropometric, ultrasonographic, and performance variables were measured, and accelerometers and three-day food diaries were provided. Within seven to 14 days, participants returned their accelerometers and their completed food diaries to the lab. Participants were between 40 and 67 years of age, not currently using any nicotine product, free of any untreated or nonresponsive diseases or conditions including neuromuscular disease or conditions that might undermine muscle health, such as diabetes, ambulatory without any assistance, and had to include both animal-based and plant-based foods in their diets. Participants were screened using the 2011 Physical Activity Readiness Questionnaire [19], a more detailed health history questionnaire, and an orthostatic hypotension test. Participants were also instructed to refrain from exercise and strenuous physical activity at least 48 h prior to the first session. The study was approved by the North Dakota State University Institutional Review Board (#HE26929 & 26153) and complied with the Helsinki Declaration of 1983. Written informed consent was obtained from all participants in this study.

### 2.1. Participant Heath Screening and Anthropometric Measures 

To screen participants for orthostatic hypotension, related to regulatory and safety concerns, resting blood pressure and standing blood pressure were measured manually with a stethoscope and Diagnostix 703 sphygmomanometer (American Diagnostic Corporation, Hauppauge, NY, USA). Those whose blood pressure dropped by more than 10 mm Hg, either systolic or diastolic, from resting to standing during the orthostatic hypotension test were excluded (*n* = 0). Following the orthostatic hypotension test, anthropometric variables were measured. Age (years) was self-reported. Height (cm) was measured using a stadiometer (Seca 213, Chino, CA, USA) and body mass (kg) was recorded using a digital balance (Denver Instrument DA-150, Arvada, CO, USA).

### 2.2. Ultrasonography

Images of the right rectus femoris muscle were captured using a Philips ultrasound system (model HD11 XE; Bothell, WA, USA) with a L12-5 50 mm linear array probe by three trained research assistants. Images were taken while participants were standing at marked sites 50% and 75% of the measured distance from the superior iliac spine of the hip to the lateral condyle of the knee. Participants were instructed to use their left leg as a base of support, while relaxing their right, resulting in a slight bend in the right knee. Previous works have shown high test–retest reliability of ultrasound measures of muscle thickness of healthy adults taken in the standing position [20,21]. A more recent study found the intraclass correlation coefficient (ICC) for standing measures of the anterior thigh muscles was 0.89, while the ICC for the same measures taken while participants were recumbent was 0.90 [22]. Following generous application of ultrasonic gel, the probe was placed on the skin perpendicular to the leg, and light, consistent pressure was applied to avoid excessive depression of the dermal surface until a full, clear image was obtained. The probe was removed from participants’ skin between each image acquisition, and markings were used to ensure the same area was measured. Because our participants were younger and likely have greater muscle size, the panoramic feature was used at the 50% site to record the entire transverse rectus femoris [10]. For panoramic ultrasonography, the lateral side of the right rectus femoris was identified, and the probe was moved medially until the entire transverse rectus femoris was recorded. B-mode image captures were taken at the 75% site where transverse sections of the rectus femoris are smaller. Three images were captured at each site using a frequency of 37 Hz with a standardized depth of 7 cm and gain of 100%. 

After each image was captured, a 1 cm line was added to each image to act as a known distance during analysis. Images were transferred to personal computers, calibrated, and analyzed. ImageJ software (National, Institutes of Health, Bethesda, MD, USA, version 1.42) was used to analyze echogenicity, cross-sectional area (CSA), and muscle thickness [23]. Echogenicity was defined as the mean pixel intensity of the rectus femoris measured in arbitrary units (A.U.) ranging between 0 (i.e., black) and 255 (i.e., white). Anatomical muscle CSA was determined by tracing the inside of the epimysium of the rectus femoris using the polygon tool. Rectus femoris thickness was assessed with a single measurement using the straight-line tool; using ImageJ, a line was made through the largest, middle portion of the muscle perpendicular to the skin. Intraclass correlation coefficients (ICC) were used to examine the reliability of these analyses. All three research assistants completed reliability training prior to being allowed to be an operator for the testing in the study. The test–retest reliability of three images obtained by the research assistants using ICCs and 95% confidence intervals were as follows: panoramic muscle thickness = 0.98 (0.90, 0.95), B-mode muscle thickness = 0.98 (0.97, 0.99), panoramic muscle area = 0.95 (0.93, 0.96), B-mode muscle area = 0.97 (0.97, 0.98), panoramic muscle echogenicity = 0.98 (0.97, 0.98), and B-mode echogenicity = 0.81 (0.75, 0.87). For consistency, these measurements were all analyzed by the same member of the research team. The mean of each participant’s values across the three images at each site (i.e., 50% and 75%) was used in our analyses. Figure 1 displays an example of muscle thickness and CSA captured and analyzed at each site.

### 2.3. Performance Measures

Participants performed a self-paced, low to moderate intensity warm-up for five minutes using a cycle ergometer. Muscle strength and endurance of the lower body were tested using isokinetic dynamometry on a Biodex Pro IV System (Biodex Medical Systems, Shirley, NY, USA). Lower body muscular strength was assessed using peak torque performed during a three-repetition test at 60° per second for knee extension–flexion and a three-repetition test at 30° per second for plantar-dorsiflexion. Lower body muscular endurance was evaluated using the total amount of work performed during a 21-repetition test at 180° per second for knee extension–flexion and 60° per second for plantar-dorsiflexion [24]. Muscular strength and then endurance were first assessed in upper leg (i.e., knee extension–flexion) and then in the lower leg (i.e., plantar-dorsiflexion). A warm-up set was completed before each lower-body strength test (i.e., knee extension–flexion and plantar-dorsiflexion); participants were instructed to perform three repetitions at ≤75% of their perceived maximal effort. Thirty seconds of rest was given between all extension-flexion tests. One minute of rest was provided between plantar-dorsiflexion tests. To optimize performance, participants were encouraged to employ “all-out effort” by research staff during all muscle function tests. To better capture muscular performance of the entire right leg, peak torques from the isokinetic strength test and total work from the isokinetic endurance test were added together to create summed peak torque and summed total work (i.e., knee extension + knee flexion + plantarflexion + dorsiflexion). 

Maximal handgrip strength (kg) was assessed using an analog Jamar Handheld Dynamometer (Bolingbrook, IL, USA). Participants were instructed to grasp the dynamometer in their dominant hand and to keep their elbow at their side with a 90° bend between the upper arm and forearm, while standing. Participants were told to squeeze the dynamometer as hard as possible for two to three seconds. Each participant performed three maximal attempts; the highest grip strength was used.

Participants then performed a 30-s chair stand test on a chair with a 43 cm floor-to-seat height. All trials were performed with participants’ arms crossed and feet at a comfortable distance apart (i.e., about hip to shoulder width). With a straight back, participants were instructed to fully sit down and stand-up for each repetition, and practice repetitions were performed to ensure adequate performance during the test. The total number of repetitions completed in 30-s period was recorded, and the 30-s period began when participants started to rise. 

### 2.4. Physical Activity Assessment 

Following performance testing, participants were given accelerometers and three-day food diaries. Physical activity was recorded using Actigraph (ActiGraph Corp, Pensacola, FL, USA) GT9X accelerometers. Participants were instructed to wear accelerometers on their right hip during all waking hours, excluding activities where the device may get wet (e.g., bathing or swimming), for a period of one week and to keep a sleep log to record the time that the accelerometer was removed at night and put back on in the morning. The raw acceleration data were collected at 80 Hz and processed in R software (http://cran.r-project.org, accessed on 6 September 2016) using the GGIR package (version 1.10-10) [25]. Non-wear time was defined as intervals of at least 90 min of zero counts with allowance of a two-minute interval of non-zero counts within a 30-min window [26]; thus, only valid time during waking hours of each day was included for statistical analyses. Although accelerometry captures many aspects of physical activity (e.g., sedentary time, light physical activity, etc.), we decided to use moderate-to-vigorous physical activity (MVPA) in our analyses because of its relationship with performance variables [27,28]. 

### 2.5. Nutrition Analysis

After performance testing, participants were also given three-day food diaries, received training on how to record dietary intakes by a member of the research team, and were required to watch a prerecorded training video. Dietary intakes from three-day food diaries, including nutritional supplements, were entered into Food Processor Nutrition Analysis Software (ESHA Research, Salem, OR, USA), which uses FoodData Central (USDA National Nutrient Database) by trained research assistants. Data entry was then line-by-line verified by a registered dietitian. Animal- and plant-based protein intakes were estimated using a line-by-line examination of dietary intake by a registered dietitian. Food items that contained less than 1 g of total protein were excluded from these calculations. Foods containing both animal- and plant-based protein were split according to their ingredients to distinguish protein sources. Animal-based protein sources included meat, fish and seafood, dairy, eggs, poultry, and wild game. 

### 2.6. Statistical Analyses

Alpha was set at 0.05, and all statistics were performed in SPSS version 27 (IBM, Armonk, NY, USA). All data are available as a Appendix A. 

#### 2.6.1. Primary Analyses: Measures from Ultrasonography and Their Relationships with Muscular Performance and the Association between Rectus Femoris Echogenicity and Specific Force

Three male participants could not be included in analyses of ultrasonography because our ultrasound machine suffered a catastrophic failure near the very end of the data collection window, precluding ultrasonography for these male participants. Thus, all analyses related to ultrasonography have 88 as opposed to 91 participants. 

We used multiple-linear regression models to determine the relationships between variables derived from ultrasonography (i.e., rectus femoris muscle thickness, echogenicity, and CSA) using the two different methodologies (i.e., panoramic versus B-mode images) and sites (i.e., 50% and 75% of right leg length) with measures of muscular performance. Each of these variables from ultrasonography were assessed in separate multiple-linear regression models. Although we consider summed peak torque and summed total work to be more representative of lower-body performance, we specifically included knee extensor peak torque and total work in these analyses, because ultrasonography was used to measure the rectus femoris, one of the knee extensors. Separate multiple-linear regression models were also used to evaluate the relationship between echogenicity and specific force of the rectus femoris, two measures of muscle quality. All aforementioned regression models were adjusted for gender (i.e., 0 = women, 1 = men), age, and body mass in kilograms divided by the square of height in meters (BMI), because these variables are routinely collected in both clinical and research settings. 

#### 2.6.2. Secondary Analyses: Animal-Based Protein Intake and Muscular Performance

All participants completed a three-day food diary, completed all performance measures (i.e., isokinetic dynamometry, handgrip strength, and 30-s chair stand test), and wore an accelerometer. For our analyses investigating nutritional variables, we first used simple linear regression models to verify that our estimates of animal-based and plant-based protein intakes together agreed with total protein intake. Animal-based and plant-based protein intakes, determined by line-by-line analysis of three-day food diaries by a registered dietitian and expressed either as relative intakes or percentages of energy intakes, were entered as predictor variables, and total protein, without partitioning into animal- or plant-based protein intakes, was the outcome variable. 

Analyses of nutritional data are complicated by the shared variance of many variables. Energy intake and macronutrient intakes, which we examined in this work, are directly related, that is, a person’s macronutrient intake, withstanding alcohol, determines their energy intake (i.e., protein + carbohydrates + fat = energy). Therefore, when analyzing dietary variables, relative energy (kcals/kg/day) and the relative intakes of all the macronutrients (g/kg/day) cannot be entered simultaneously. We used Pearson Product–Moment Coefficients to examine the collinearity of both relative macronutrient intakes and macronutrient intakes as percentages of energy intake with one another and with relative energy intake. Although there are other methodologies, we chose to include relative energy intake (kcal/kg/day) in our analyses and to express the intake of the macronutrients as percentages of energy intake. This method allowed us to control for both relative energy intake and macronutrient intakes in our statistical models. 

Mixed linear models were used to evaluate the impact of ABPI on muscular performance. The 41 men and 50 women were first blocked according to self-reported gender (0 = women, 1 = men). Then, each gender was split at their median of energy intake from animal-based protein. More specifically, gender and ABPI (below median = 0, above median = 1) were entered as fixed factors. Age, BMI, MVPA, relative energy intake, and percent energy from protein, fat, and carbohydrates were entered as continuous covariates. Models were evaluated for equality of error of variance using Levene’s Test of Equality of Variance and for heteroscedasticity using White’s Test of Heteroscedasticity; mixed models that were significantly unequal in their variances or heteroscedastic were transformed using the square root function. Out of an abundance of caution, we chose to use the HC3 method to calculate the standard errors of our variables, as it is more robust to unequal variances, heteroscedasticity, and multicollinearity than the ordinary least squares method [29]. We did not hypothesize that there would be interaction between gender and ABPI, so only main effects were examined in these mixed models. For those models in which ABPI is significant, we evaluated effect size using partial eat squared. We also sought to verify that ABPI and not total protein intake is important to performance. We verified our results by performing the same aforementioned methods, but we split each gender at median of total protein intake as a percentage of energy intake and included ABPI as a percentage of energy intake as a continuous covariate. 

Estimates of physical activity from accelerometry are considered valid when the devices are worn for 10 h per day for at least four days [28], and three participants failed to meet these criteria despite our instruction to wear the devices during all waking hours for one week. Nonetheless, all other participants achieved at least four or more days including one weekend day with an average of 10 or more hours of time wearing the device. These three participants who failed to wear accelerometers as directed represent a small portion of our sample (3.3%), and physical activity was included in our mixed models as a covariate; physical activity is not the focus of this work, but we feel it is essential to control for in our mixed models evaluating ABPI. For these reasons and due to small sample size, particularly when split into groups, we decided to include these three participants, using their limited physical activity data in our analyses. 

#### 2.6.3. Descriptive Statistics 

For our descriptive statistics, we described the four groups from the secondary analyses in our all of our tables, even though we chose not to investigate the association between ABPI and measures from ultrasonography, because the three men who were precluded from ultrasonography were, coincidently, above the median for animal-based protein intake as a percentage of energy. Within these tables, we chose to use the Brown–Forsythe method for comparisons, because we did not assume equal variances. We compared those above the median of ABPI as a percentage of energy to those below the median within each gender, so we did not adjust for multiple comparisons.

## 3. Results

Table 1 describes participants self-reported age, measured height, weight, and calculated BMI. There were no statistically significant differences between those below or above the median of ABPI as a percentage of total energy within each gender.

Table 2 describes right rectus femoris muscle thickness, echogenicity, and CSA measured using the panoramic ultrasonography at 50% and B-mode images at 75% of the distance of the right leg. Within each gender, there were no statistically significant differences in these measures between those above the median of ABPI and those below.

Table 3 presents the results of the sperate multiple-linear regression models investigating the relationship between different measures derived from ultrasonography and muscular performance. Measures of rectus femoris size assessed using panoramic ultrasonography were less related to knee extensor performance but more strongly related to overall muscular performance. More specifically, both muscle thickness (*p* = 0.302) and CSA (*p* = 0.056) assessed using the panoramic feature of the right leg were unrelated to knee extensor peak torque, whereas the same measures assessed using a B-mode image at of the right leg were related to knee extensor peak torque. Similarly, muscle thickness assessed using the panoramic feature was unrelated to knee extensor total work (*p* = 0.197). Although muscle CSA captured with the panoramic feature was related to knee extensor total work (*p* = 0.049), it was less closely related than muscle CSA (*p* = 0.013) or thickness (*p* = 0.036) assessed with a B-mode image at 75% of leg length. Conversely, measures of muscle thickness (*p* = 0.001) and CSA (*p* = 0.004) derived from panoramic ultrasound were significantly related to handgrip strength performance, whereas the same measures collected using B-mode were not. Muscle CSA from panoramic ultrasound was also most closely related to summed peaked torque (*p* = 0.005), a relationship that was only close to significance (*p* = 0.051) with a B-mode image. Both methodologies (i.e., panoramic and B-mode) produced measures of muscle thickness and CSA that were associated with summed total work. 

Echogenicity of rectus femoris was unrelated to both knee extensor and summed peak torque but was significantly associated with knee extensor total work when captured using either panoramic (*p* = 0.001) or B-mode images (*p* = 0.004). Echogenicity of the rectus femoris from both panoramic (*p* = 0.008) and B-mode (*p* = 0.007) images was also associated with handgrip strength. Interestingly, although echogenicity was related to knee extensor total work, it was not related to summed total work when using either methodology. No ultrasonographic measure was associated with 30-s chair stand performance.

Table 4 describes our evaluation of echogenicity with specific force, two measures of muscle quality. Echogenicity was not related to specific force in any regression model nor was any model significant. We found measures from the 50% site, taken using the panoramic feature, created better fitting models. In fact, echogenicity assessed at 50% trended toward significance (*p* = 0.077).

Table 5 describes the nutritional variables assessed from three-day food diaries for study participants. There were significant differences in macronutrient intake between those above the median for ABPI as a percentage of energy intake and those below within each gender; relative carbohydrate intake, carbohydrate intake as percentage of energy, protein intake as percentage of energy, relative ABPI, ABPI as a percentage of energy, and relative plant-based protein intake were all significantly different in both men and women. Those above the median consumed less carbohydrates, more protein, and more animal based protein than those below. In women, there were also significant differences in relative fat and calcium intake with those above the median consuming less fat and more calcium. In men, on the other hand, there was a significant difference in relative energy intake with those below the median of ABPI consuming more energy.

Table 6 lists physical activity variables recorded using accelerometry. Excluding wear days, which were greater in men below the median compared to men above the median, there were no significant differences between those above the median of animal-based protein as percentage of energy intake and those below. 

Regression models examining estimates of animal-based and plant-based protein intakes with total protein intake showed good agreement between our estimates and total protein. Estimates of relative animal-based and relative plant-based protein intakes explained 98.4% of the variance in relative protein intake (F2,88 = 2788.702, *p* < 0.001), and estimates of animal- and plant-based protein intakes as percentages of energy explained 94.0% of the variance in protein as a percentage of energy (F2,88 = 683.550, *p* < 0.001). 

Table 7 shows Pearson Product–Moment Correlations between relative macronutrient intakes, macronutrient intakes as percentages of energy intake, and relative energy intake. Relative macronutrient intakes showed stronger relationships with relative energy intake than macronutrient intakes expressed as a percentage of energy intake. Withstanding the association between percent of energy from fats and carbohydrates, macronutrient intakes expressed as percentages of energy were less strongly correlated amongst one another than relative macronutrient intakes. These results suggest macronutrient intakes should be expressed as percentages of energy intake in statistical models including relative energy intake to limit collinearity. 

Table 8 and Figure 2 present the results of our investigation of the relationship between ABPI with performance measures. To create homoscedastic models with equal variances, data from the handgrip strength test (kg) and the 30-s chair stand test (repetitions) were transformed using the square root function. Using these transformed variables, all of these mixed models had equal variances according to Levene’s Test and were homoscedastic according to White’s test (i.e., *p* > 0.05). 

Our mixed models explained 78.6% of the variance of summed peak torque performed during the isokinetic strength test, 75.7% of the variance of summed work performed during the isokinetic endurance test, and 83.3% of the variance in handgrip strength transformed using the square root function, indicating good model fit for these performance variables. However, our mixed model investigating the results of the 30-s chair stand test only explained 19.1% of the variance in this measure, indicating relatively poor model fit. Nonetheless, all models were significant. 

Animal-based protein intake was significant to mixed models evaluating lower-body muscular strength, lower-body muscular endurance, and handgrip strength. Those consuming above the median of animal-based protein as percentage of energy intake performed better on these tests of muscular strength and endurance than those below the median. The effect sizes assessed using partial eta squared of the ABPI median split were 0.120, 0.065, and 0.049 for summed lower-body peak torque, summed lower-body total work, and handgrip strength, respectively. Animal-based protein intake was not related to performance in the 30-s chair stand test. 

Because ABPI was significant to lower-body muscular strength, lower-body muscular endurance, and handgrip strength, we wanted to verify that these findings were due to ABPI and not to greater total protein intake. Although we did control for total protein intake as percentage of energy in our mixed models where participants were split at the median of ABPI, Table 9 shows our analyses where participants were split at the median of total protein intake as percentage of energy intake and ABPI as a percent of energy intake was entered as a continuous covariate. With the exception of square root transformed 30-s chair stand repetitions, all of these mixed models had equal variances according to Levene’s Test and were homoscedastic according to White’s test (i.e., *p* > 0.05). Square root transformed 30-s chair stand performance was homoscedastic but showed unequal variances between groups (*p* = 0.024) according to Levene’s test. Because our earlier analysis of square root transformed 30-s chair stand performance (i.e., Table 8) showed equal variances between groups, was homoscedastic, and produced nonsignificant results regarding protein intake and ABPI, we did not transform 30-s chair stand performance using a different methodology (e.g., Log). In other words, square root transformed 30-s chair stand performance was included in Table 9 despite showing unequal variances between groups, although the HC3 method is considered to be more robust to violations of unequal variance [28]. Total protein intake split at the median of energy intake was not significant to any performance variable, whereas APBI split at the median was significant to lower-body muscular strength, lower-body muscular endurance, and handgrip strength, indicating that APBI is more closely related to muscular performance than total protein intake. 

## 4. Discussion

We found that measures of muscle size from standardized B-mode ultrasound images better captured the performance of the knee extensors, whereas measures of muscle size assessed from panoramic images were more closely related to overall muscular performance, producing significant associations between muscle size with summed peak torque and handgrip strength. However, our methodology differed from that of others who have utilized panoramic ultrasound. We took panoramic images of the rectus femoris at one location (i.e., 50% of leg length) as opposed to using a template to image the entire length of the quadriceps, although one research group advocated for an investigation of a single site at the mid-quadriceps [11].

Nonetheless, the lack of a significant relationship between muscle thickness and CSA measured using the panoramic feature and knee extensor strength is surprising, considering these measures of muscle size were more closely related both to lower-body strength (i.e., summed peak torque) and upper-body strength. Low muscle strength is the first criterion of sarcopenia according to the European Working Group on Sarcopenia in Older People 2 and should be, albeit not necessarily linearly, related to muscle mass [1]. In other words, changes in muscle mass or size are not as meaningful as changes in muscle strength. Measures of muscle size or mass that are unrelated to muscle strength then may have limited utility in assessing or screening for sarcopenia. Despite the fact measures from panoramic ultrasonography lacked face validity in the form of a significant relationship with knee extensor peak torque, our findings suggest that the panoramic feature is a suitable method for assessing sarcopenia in those with greater muscle at the midpoint of thigh, as it is related to both lower-body and upper-body strength. 

We also report that in our sample echogenicity was unrelated to both knee extensor, strength, overall lower-body strength, and rectus femoris specific force, another measure of muscle quality. Although Strasser and colleagues [13] reported a significant correlation between echogenicity and knee extensor strength, the relationship was only found in younger and not older adults. In contrast, Akima and colleagues [30] found a significant relationship between echogenicity and sit-to-stand performance in older Japanese men and women. However, in a subsequent work, the same research group reported no relationship between echogenicity and knee extensor strength [6]. We also did not find a significant relationship between echogenicity and knee extensor strength, and we were the first, at least to our knowledge, to directly compare the echogenicity of the rectus femoris to the muscle’s specific force. None of the relationships were significant. However, we did find an association between echogenicity with handgrip strength and knee extensor muscular endurance. Echogenicity has been related to both intramuscular fat [31] and fibrous tissue [32] content of muscle. In a large study of older Italian men and women, De Stefano and colleagues [33] reported a negative association between intramuscular fat and physical performance but found that those who were overweight or “Class I” obese had greater knee extensor strength than those with a normal BMI, suggesting that intramuscular fat plays a greater role in physical performance than in maximal strength. Our findings regarding echogenicity support that view. Echogenicity, then, is not closely related to specific force as it is with other muscular qualities such as endurance, because specific force is dependent on maximal muscle strength. 

Our secondary findings regarding dietary intake indicate a positive relationship between ABPI and muscle strength when controlling for gender, age, BMI, relative energy intake, and macronutrient composition. More specifically, those above the median of ABPI as percentage of energy intake showed greater lower-body strength and endurance and greater handgrip strength than those below. Although greater protein intake is thought to be protective from developing sarcopenia [34,35,36], a recent cross-sectional study of older Danish adults utilizing methods similar to ours (e.g., three-day food diary and physical activity assessment) reported that protein intake was not related to knee extensor strength, handgrip strength, and 30-s chair stand test performance [16]. In contrast to their methodology where participants were divided into groups based on relative protein intake, we split ours according to ABPI as a percentage of energy intake. Although recommendations for protein intake are made on a g/kg basis [36], an advantage of expressing intakes as percentages of energy intake is that one can control for relative energy intakes and for macronutrient composition in the same statistical model. There is a high degree of collinearity between relative intakes of macronutrients and relative energy intake. In fact, one of the main findings from Højfeldt and colleagues’ study of older Danish adults was that relative protein intakes and relative energy intakes are related [16]. Collinearity can bias estimates of betas in multivariate analyses [37]. Although there is still a degree of collinearity between macronutrient intakes as percentages of energy and relative energy intakes, we addressed this issue by using the HC3 method of calculating standard errors, which is more robust to collinearity and heteroscedasticity [29]. Outside of expressing intakes as percentages of energy, our methodology also differed because we evaluated ABPI. Plant-based proteins generally contain amino acids that are oxidized to be used as energy to a greater extent than higher quality animal-based proteins [18]. Thus, total protein intake is likely less strongly related to muscle mass and strength than protein intake from higher quality sources, and our findings particularly support this notion. When split at its median, total protein intake as a percentage of energy intake was not related to lower-body strength, lower-body endurance, and handgrip strength, whereas ABPI split at the median was positively associated with all these measures. 

There are some limitations to our investigations. We cannot determine from our primary results if the panoramic feature inaccurately quantified muscle size, because our study lacked a measure of criterion validity in the form rectus femoris muscle thickness and cross-sectional area assessed using MRI or CT. Another caveat to our findings regarding ultrasonography is the skill of our sonographers. Although our sonographers were trained and showed good reliability, ICCs were greater than 0.95 for all measures other than B-mode echo intensity, which was equal to 0.81; they were and are not professional sonographers. Panoramic ultrasound is a more difficult method to perform, as the probe must be moved while keeping light, consistent pressure during imaging. Our results regarding panoramic ultrasonography and knee extensor performance may indicate, then, that the method should only be performed by those with highest levels of skill. Nonetheless, measures from panoramic ultrasonography were related to summed peak torque and handgrip strength, indicating these measures were related to overall performance. Another potential limitation was the assessment of anatomical as opposed to physiological CSA, as physiological CSA of pennate muscles, such as the rectus femoris, is thought to be more closely related to strength [10].

Regarding the limitations of our secondary analysis, this was a cross-sectional study incapable of establishing causality, the self-reported nature of our food-diary recording limits its accuracy, and we included three participants’ physical activity data despite the fact these participants did not have enough valid wear days. Our secondary investigation did have some strengths. We objectively measured and controlled for physical activity. We verified our partitioning of protein intake into animal- and plant-based sources using regression models. We included relative energy and macronutrient intakes in our mixed models to control for differences in participants’ diets outside of ABPI. Lastly, we confirmed the importance of ABPI to muscular performance by performing another set on analyses where participants were spilt at the median of percent energy from total protein. 

## 5. Conclusions

We report that measures of muscle thickness and CSA derived from panoramic ultrasonography are more closely related to overall strength than the same measures derived from B-mode ultrasound images. Thus, panoramic images may be a suitable method to measure muscle size and estimate overall muscle mass when the entire transverse area of a muscle cannot be measured with a standardized B-mode image. However, measures of muscle size from B-mode images were more closely related to the performance of knee extensors alone, suggesting that B-mode images may be better measures of individual muscles or muscle groups. Echogenicity of the rectus femoris was unrelated to its specific force and to overall lower-body strength. Instead, echogenicity was related to handgrip strength and knee extensor endurance. Finally, we found a positive relationship between ABPI and lower-body strength, lower-body endurance, and handgrip strength when controlling for physical activity and diet. 

## Figures and Tables

**Figure 1 jcm-10-00988-f001:**
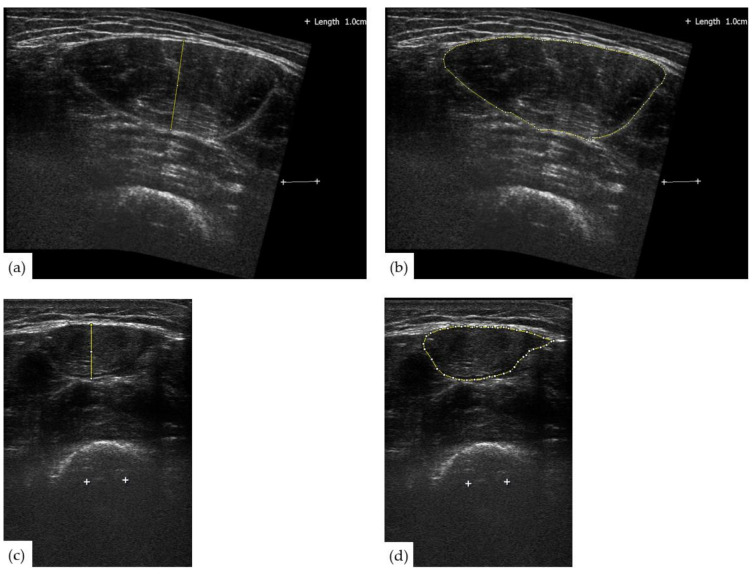
Examples of rectus femoris muscle thickness and CSA captured via ultrasonography for one participant. (**a**) Rectus femoris muscle thickness at 50% of leg length captured using the panoramic feature. (**b**) Same as A but showing muscle CSA. (**c**) Rectus femoris muscle thickness at 75% of leg length captured using a standardized B-mode image. (**d**) Same as C but showing muscle CSA.

**Figure 2 jcm-10-00988-f002:**
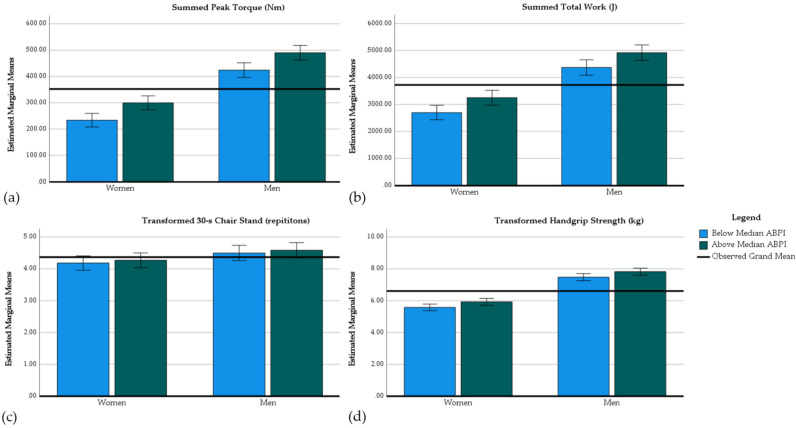
Animal-based protein intake and muscular performance. Animal-based protein intake was split at the median of percent energy from animal-based protein within both men and women; below median = 0, above median = 1. Covariates included age, gender, BMI, MVPA, relative energy intake, and percentages of energy intake from fat, carbohydrate, and protein. All bars are means, and error bars represent 95% confidence intervals. (**a**) Summed isokinetic peak torque by gender and animal-based protein intake. Summed isokinetic peak torque was calculated by adding the peak torques recorded during the isokinetic strength test, 60° per second for knee extension–flexion and 30° per second for plantar-dorsiflexion. (**b**) Summed isokinetic endurance by gender and animal-based protein intake. Summed isokinetic endurance was calculated by adding total work performed during a 21-repetition test at 180° per second for the knee extension–flexion and 60° per second for plantar-dorsiflexion. (**c**) Square root transformed 30-s chair stand test repetitions by gender and animal-based protein intake. The height of the chair for the 30-s chair stand test was 43 cm. (**d**) Square root transformed handgrip strength by gender and animal-based protein intake.

**Table 1 jcm-10-00988-t001:** Self-reported age and anthropometrics for 41 men and 50 women.

	Women			Men		
	Total (*n* = 50)	Below Median ABPI (*n* = 25)	Above Median ABPI (*n* =25)	Total (*n* = 41)	Below Median ABPI (*n* = 21)	Above Median ABPI (*n* =20)
Age (years)	54.00	55.00	54.00	51.00	55.00	50.00
Height (cm)	165.20	164.00	165.50	181.00	176.70	181.05
Weight (kg)	68.30	67.33	69.12	87.7	85.20	92.36
BMI	25.11	24.43	25.54	26.57	26.57	26.32

All values are medians. Comparisons within gender and between those below and above the median for animal-based protein intake (ABPI) as a percentage of energy intake were made using the Brown–Forsythe method.

**Table 2 jcm-10-00988-t002:** Rectus femoris muscle thickness, echogenicity, and cross-sectional area assessed via ultrasonography captured using the panoramic feature at 50% and with regular B-mode images at 75% of the right leg in 88 middle-aged men and women.

	Women			Men		
	Total (*n* = 50)	Below Median ABPI (*n* = 25)	Above Median ABPI (*n* =25)	Total (*n* = 38)	Below Median ABPI (*n* = 21)	Above Median ABPI (*n* =17)
Muscle Thickness at 50% (cm)	2.109	2.038	2.178	2.339	2.275	2.345
Muscle Thickness at 75% (cm)	0.707	0.710	0.706	0.994	0.918	1.070
Echogenicity at 50% (A.U.)	96.70	97.86	96.64	35.90	34.85	41.73
Echogenicity at 75% (A.U.)	91.99	93.34	90.63	81.99	74.56	84.54
Muscle CSA at 50% (cm^2^)	7.384	6.569	7.861	10.593	10.470	10.963
Muscle CSA at 75% (cm^2^)	0.957	0.790	1.055	1.934	1.660	2.088

All values are medians. CSA = Muscle Cross-Sectional Area. A.U. = Arbitrary Units. Comparisons within gender and between those below and above the median for animal-based protein intake (ABPI) as a percentage of energy intake were made using the Brown–Forsythe method.

**Table 3 jcm-10-00988-t003:** The associations between different ultrasonographic measures of the right rectus femoris using the panoramic feature (50% of leg upper length) and a B-mode image (75% of upper leg length) in a sample of 88 middle-aged men and women when controlling for age, gender, and BMI.

Variable Entered	Dependent Variable											
	Knee Extensor Peak Torque (Nm)		Summed Peak Torque (Nm)		Knee Extensor Total Work (J)		Summed Total Work (J)		30-s Chair Stand Test (Repetitions)		Handgrip Strength (kg)	
	R	B ± S.E.	R	B ± S.E.	R	B ± S.E.	R	B ± S.E.	R	B ± S.E.	R	B ± S.E.
Muscle Thickness at 50% (cm)	0.816 *p* < 0.001	11.098 ± 10.286 *p* = 0.302	0.861 *p* < 0.001	42.622 ± 20.024 *p* = 0.036	0.707 *p* < 0.001	174.654 ± 134.410 *p* = 0.197	0.850 *p* < 0.001	595.980 ± 200.221 *p* = 0.004	0.353 *p* = 0.025	1.348 ± 1.415 *p* = 0.334	0.900 *p* < 0.001	6.966 ± 3.328 *p* = 0.001
Muscle Thickness at 75% (cm)	0.826 *p* < 0.001	23.166 ± 9.955 *p* = 0.022	0.862 *p* < 0.001	42.533 ± 19.076 *p* = 0.025	0.719 *p* < 0.001	269.252 ± 126.430 *p* = 0.036	0.849 *p* < 0.001	555.550 ± 191.981 *p* = 0.005	0.347 *p* = 0.029	0.963 ± 1.357 *p* = 0.480	0.885 *p* < 0.001	0.307 ± 2.131 *p* = 0.886
Echogenicity at 50% (A.U.)	0.822 *p* < 0.001	−0.271 ± 0.141 *p* = 0.059	0.854 *p* < 0.001	−0.237 ± 0.275 *p* = 0.389	0.854 *p* < 0.001	−5.809 ± 1.710 *p* = 0.001	0.836 *p* < 0.001	−3.622 ± 2.804 *p* = 0.200	0.349 *p* = 0.027	−0.016 ± 0.019 *p* = 0.412	0.895 *p* < 0.001	−0.078 ± 0.029 *p* = 0.008
Echogenicity at 75% (A.U.)	0.817 *p* < 0.001	−0.142 ± 0.129 *p* = 0.274	0.853 *p* < 0.001	−0.058 ± 0.248 *p* = 0.815	0.853 *p* < 0.001	−4.763 ± 1.550 *p* = 0.003	0.834 *p* < 0.001	−4.763 ± 1.550 *p* = 0.370	0.376 *p* = 0.012	−0.027 ± 0.017 *p* = 0.113	0.895 *p* < 0.001	−0.071 ± 0.026 *p* = 0.007
Muscle CSA at 50% (cm^2^)	0.823 *p* < 0.001	3.406 ± 1.754 *p* = 0.056	0.867 *p* < 0.001	9.915 ± 3.271 *p* = 0.005	0.717 *p* < 0.001	44.281 ± 22.142 *p* = 0.049	0.860 *p* < 0.001	126.648 ± 32.205 *p* < 0.001	0.349 *p* = 0.028	0.193 ± 0.237 *p* = 0.418	0.897 *p* < 0.001	1.050 ± 0.354 *p* = 0.004
Muscle CSA at 75% (cm^2^)	0.828 *p* < 0.001	8.120 ± 3.245 *p* = 0.014	0.860 *p* < 0.001	12.464 ± 6.294 *p* = 0.051	0.726 *p* < 0.001	104.435 ± 40.951 *p* = 0.013	0.844 *p* < 0.001	153.621 ± 63.783 *p* = 0.018	0.341 *p* = 0.034	0.165 ± 0.445 *p* = 0.713	0.885 *p* < 0.001	−0.154 ± 0.698 *p* = 0.826

S.E. = standard error. Age: years. Gender: Women = 0, Men = 1; CSA = Muscle Cross-Sectional Area; BMI: kg/m^2^. Summed peak torque was calculated by adding the peak torques recorded during the isokinetic strength test, 60° per second for knee extension-flexion and 30° per second for plantar-dorsiflexion. Summed isokinetic endurance was calculated by adding total work performed during a 21-repetition test at 180° per second for the knee extension–flexion and 60° per second for plantar-dorsiflexion. The height of the chair for the 30-s chair stand test was 43 cm.

**Table 4 jcm-10-00988-t004:** Association of echogenicity assessed via ultrasonography captured using the panoramic feature and B-mode images of the right leg with various assessments of knee extensor specific force in 88 middle-aged men and women.

Variable Entered	Specific Force Variable	R	F_4,83_	Age (Beta ± S.E.)	Gender (Beta ± S.E.)	BMI (Beta ± S.E.)	Entered Variable (Beta ± S.E.)
Echogenicity at 50% (A.U.)	Peak KE Torque by Muscle Thickness at 50% (Nm/cm)	0.299	2.030 *p* = 0.098	−0.799 ± 3.154 *p* = 0.801	−106.185 ± 54.253 *p* = 0.054	10.527±4.306 *p* = 0.017	−1.381 ± 0.770 *p* = 0.077
	Peak KE Torque by Muscle CSA at 50% (Nm/cm^2^)	0.311	2.226 *p* = 0.073	−0.625 ± 2.187 *p* = 0.776	−5.110 ± 37.627 *p* = 0.892	6.163±2.986 *p* = 0.042	−0.831 ± 0.534 *p* = 0.123
Echogenicity at 75% (A.U.)	Peak KE Torque by Muscle Thickness at 75% (Nm/cm)	0.239	1.253 *p* = 0.295	−0.074 ± 3.181 *p* = 0.982	−45.255 ± 41.943 *p* = 0.284	9.403±4.341 *p* = 0.033	−0.370 ± 0.702 *p* = 0.600
	Peak KE Torque by Muscle CSA at 75% (Nm/cm^2^)	0.267	1.594 *p* = 0.184	−0.161 ± 2.199 *p* = 0.535	32.388 ± 28.991 *p* = 0.267	5.416±3.001 *p* = 0.075	−0.131 ± 0.485 *p* = 0.788

A.U. = Arbitrary Units. S.E. = Standard Error. Age: years. Gender: Women = 0; Men = 1. BMI: kg/m^2^.

**Table 5 jcm-10-00988-t005:** Dietary intakes accessed from three-day food diaries in 41 middle-aged men and 50 middle-aged women.

	Women			Men		
	Total (*n* = 50)	Below Median ABPI (*n* = 25)	Above Median ABPI (*n* = 25)	Total (*n* = 41)	Below Median ABPI (*n* = 21)	Above Median ABPI (*n* = 20)
Relative Energy (kcal/kg/day)	24.46	30.51	22.51	28.41	31.08 *	26.73
Relative Fat (g/kg/day)	1.04	1.14 *	0.90	1.15	1.20	0.99
Fat Percent Energy (%)	35.66	37.03	34.88	34.85	34.02	35.63
Relative Carbohydrate (g/kg/day)	2.85	3.22 **	2.30	3.56	4.12 **	2.81
Carbohydrate Percent Energy (%)	46.20	48.56 *	44.36	46.86	48.82 ***	41.16
Relative Protein (g/kg/day)	1.19	1.15 *	1.25	1.28	1.28	1.24
Protein Percent Energy (%)	17.99	14.40 **	21.27	17.35	14.54 ***	18.65
Relative Animal Protein (g/kg/day)	0.77	0.61 ***	1.00	0.87	0.82 *	0.96
Animal Protein Percent Energy (%)	11.99	8.59 ***	16.08	11.74	10.39 ***	15.16
Relative Plant Protein (g/kg/day)	0.31	0.37*	0.27	0.34	0.39 **	0.29
Plant Protein Percent Energy (%)	4.92	5.23	4.81	4.56	4.77	4.26
Vitamin D (IU/day)	155.28	105.58	236.41	149.70	206.52	135.49
Calcium (mg/day)	849.06	743.91 **	951.94	1166.69	1103.57	1212.28
Mg (mg/day)	202.96	196.17	210.15	315.96	254.04	332.94
Mn (mg/day)	1.67	1.50	1.98	2.03	2.31	1.89
Vitamin K (mcg/day)	72.01	88.31	59.97	70.72	52.02	77.98
Fe (mg/day)	12.49	12.51	12.03	16.10	18.43	14.80
Vitamin C (mg/day)	107.42	84.78	115.31	79.03	86.42	54.11
Vitamin E (mg/day)	7.716	7.00	13.06	7.71	5.37	8.10
P (mg/day)	772.54	809.96	765.45	1314.39	1265.21	1349.81
K (mg/day)	1693.39	1692.27	1754.97	2577.01	2577.01	2576.71

All values are medians. Comparisons between those below and above the median for animal-based protein intake (ABPI) as a percentage of energy intake within gender were made using the Brown–Forsythe method. * *p* < 0.05; ** *p* < 0.01; *** *p* < 0.001.

**Table 6 jcm-10-00988-t006:** Physical activity variables assessed using accelerometry in 41 middle-aged men and 50 middle-aged women.

	Women			Men		
	Total (*n* = 50)	Below Median ABPI (*n* = 25)	Above Median ABPI (*n* = 25)	Total (*n* = 41)	Below Median ABPI (*n* = 21)	Above Median ABPI (*n* = 20)
Wear Days (days)	7.00	6.00	7.007	7.00	7.00 *	6.00
Wear Time (min/day)	867.04	869.50	864.57	895.33	895.71	891.87
Sedentary Time (min/day)	559.58	556.00	563.001	613.14	606.00	620.91
Light Physical Activity (min/day)	265.13	285.83	260.33	242.38	269.43	210.11
Moderate Physical Activity (min/day)	27.46	30.67	22.00	27.86	31.83	25.85
Vigorous Physical Activity (min/day)	0.15	0.14	0.29	0.33	2.00	0.00
Moderate to Vigorous Physical Activity (min/day)	31.05	31.20	27.14	33.25	33.83	27.00

All values are medians. Comparisons between those below and above the median for animal-based protein intake (ABPI) as a percentage of energy intake within gender were made using the Brown–Forsythe method. * *p* < 0.05.

**Table 7 jcm-10-00988-t007:** Pearson Product–Moment Correlations of macronutrient intakes, including animal-based protein, and relative energy intake in 41 middle-aged men and 50 middle-aged women.

Variable	Variable							
	Relative Energy Intake	Relative Fat (g/kg/day)	Fat Percent Energy (%)	Relative Carbohydrate (g/kg/day)	Carbohydrate Percent Energy (%)	Relative Protein (g/kg/day)	Protein Percent Energy (%)	Relative Animal Protein (g/kg/day)
Relative Fat (g/kg/day)	0.819 *p* < 0.001	-	-	-	-	-	-	-
Fat Percent Energy (%)	−0.120 *p* = 0.258	0.435 *p* < 0.001	-	-	-	-	-	-
Relative Carbohydrate (g/kg/day)	0.911 *p* < 0.001	0.534 *p* < 0.001	-0.440 *p* < 0.001	-	-	-	-	-
Carbohydrate Percent Energy (%)	0.315 *p* = 0.002	−0.188 *p* = 0.074	−0.845 *p* < 0.001	0.648 *p* < 0.001	-	-	-	-
Relative Protein (g/kg/day)	0.755 *p* < 0.001	0.617 *p* < 0.001	−0.144 *p* = 0.174	0.570 *p* < 0.001	−0.019 *p* = 0.858	-	-	-
Protein Percent Energy (%)	−0.353 *p* = 0.001	−0.351 *p* = 0.001	−0.114 *p* = 0.281	−0.438 *p* < 0.001	−0.438 *p* < 0.001	0.297 *p* = 0.004	-	-
Relative Animal Protein (g/kg/day)	0.548 *p* < 0.001	0.452 *p* < 0.001	−0.122 *p* = 0.248	0.357 *p* = 0.001	−0.138 *p* = 0.191	0.922 *p* < 0.001	0.473 *p* < 0.001	-
Animal Protein Percent Energy (%)	−0.350 *p* < 0.001	−0.332 *p* = 0.001	−0.082 *p* = 0.439	−0.440 *p* < 0.001	−0.431 *p* < 0.001	0.277 *p* = 0.008	0.916 *p* < 0.001	0.550 *p* < 0.001

**Table 8 jcm-10-00988-t008:** Animal-based protein intake and muscular performance in middle-aged men and women.

Performance Variable	R	F_9,81_	Age (Beta ± S.E.)	Gender (Beta ± S.E.)	BMI (Beta ± S.E.)	MVPA (Beta ± S.E.)	Relative Energy (Beta ± S.E.)	Fat Percent Energy (Beta ± S.E.)	Carbohydrate Percent Energy (Beta ± S.E.)	Protein Percent Energy (Beta ± S.E.)	Animal-Based Protein Intake Median Split (Beta ± S.E.)
Summed Isokinetic Peak Torque (Nm)	0.887	33.111 *p* < 0.001	−3.767 ± 1.138 *p* = 0.001	190.543 ± 13.850 *p* < 0.001	1.694 ± 1.874 *p* = 0.369	0.287 ± 0.395 *p* = 0.469	−0.829 ± 0.862 *p* = 0.339	−3.754 ± 8.467 *p* = 0.659	−3.889 ± 8.351 *p* = 0.643	−5.769 ± 8.007 *p* = 0.473	65.874 ± 19.855 *p* = 0.001
Summed Isokinetic Work (J)	0.870	28.032 *p* < 0.001	−46.224 ± 11.546 *p* < 0.001	1671.298 ± 126.695 *p* < 0.001	29.436 ± 19.814 *p* = 0.141	2.842 ± 4.617 *p* = 0.540	16.825 ± 9.500 *p* = 0.080	−100.977 ± 76.033 *p* = 0.188	−95.794 ± 76.033 *p* = 0.204	−92.620 ± 71.011 *p* = 0.196	549.944 ± 232.478 *p* = 0.020
Transformed 30-Second Chair Stand (repetitions #)	0.437	2.128 *p* = 0.036	0.004 ± 0.010 *p* = 0.700	0.316 ± 0.128 *p* = 0.016	−0.024 ± 0.013 *p* = 0.081	0.000 ± 0.003 *p* = 0.940	0.008 ± 0.009 *p* = 0.859	−0.092 ± 0.077 *p* = 0.237	−0.103 ± 0.076 *p* = 0.182	−0.095 ± 0.076 *p* = 0.214	0.086 ± 0.156 *p* = 0.584
Transformed Handgrip Strength (kg)	0.913	45.026 *p* < 0.001	−0.029 ± 0.008 *p* = 0.001	1.898 ± 0.105 *p* < 0.001	0.001 ± 0.018 *p* = 0.956	0.003 ± 0.003 *p* = 0.295	−0.008 ± 0.008 *p* = 0.323	−0.083 ± 0.042 *p* = 0.052	−0.091 ± 0.041 *p* = 0.027	−0.111 ± 0.040 *p* = 0.007	0.349 ± 0.171 *p* = 0.045

S.E. = standard error. Age: years. Gender: Women = 0, Men = 1. BMI: kg/m^2^. Relative energy intake: kcal/kg/day. Animal-based protein intake was split at the median of percent energy from animal-based protein within both men and women; below median = 0, above median = 1. Nutritional variables were assessed using three-day food diaries. Summed isokinetic peak torque was calculated by adding the peak torques recorded during the isokinetic strength test, 60° per second for knee extension–flexion and 30° per second for plantar-dorsiflexion. Summed isokinetic endurance was calculated by adding total work performed during a 21-repetition test at 180° per second for the knee extension–flexion and 60° per second for plantar-dorsiflexion. Total repetitions performed during the 30-s chair stand test and handgrip strength were transformed using the square root function. The height of the chair for the 30-s chair stand test was 43 cm.

**Table 9 jcm-10-00988-t009:** Total protein intake and muscular performance in middle-aged men and women.

Performance Variable	R	F_9,81_	Age (Beta ± S.E.)	Gender (Beta ± S.E.)	BMI (Beta ± S.E.)	MVPA (Beta ± S.E.)	Relative Energy (Beta ± S.E.)	Fat Percent Energy (Beta ± S.E.)	Carbohydrate Percent Energy (Beta ± S.E.)	ABPI Energy (Beta ± S.E.)	Total Protein Intake Median Split (Beta ± S.E.)
Summed Isokinetic Peak Torque (Nm)	0.871	28.366 *p* < 0.001	−4.013 ± 1.171 *p* = 0.001	189.571 ± 14.575 *p* < 0.001	2.003 ± 2.029 *p* = 0.326	0.194 ± 0.427 *p* = 0.651	−0.792 ± 0.962 *p* = 0.413	−0.049 ± 4.836 *p* = 0.992	−0.681 ± 4.576 *p* = 0.882	1.754 ± 3.637 *p* = 0.631	19.397 ± 23.176 *p* = 0.405
Summed Isokinetic Work (J)	0.856	24.638 *p* < 0.001	−47.751 ± 12.387 *p* < 0.001	1654.781 ± 134.463 *p* < 0.001	32.111 ± 22.256 *p* = 0.153	2.090 ± 5.296 *p* = 0.694	16.687 ± 10.609 *p* = 0.120	−24.735 ± 61.303 *p* = 0.688	−24.971 ± 58.990 *p* = 0.673	29.836 ± 43.397 *p* = 0.494	−2.405± 258.849 *p* = 0.993
Transformed 30-Second Chair Stand (repetitions #)	0.409	1.806 *p* = 0.080	0.004 ± 0.011 *p* = 0.728	0.313 ± 0.130 *p* = 0.018	−0.024 ± 0.013 *p* = 0.084	0.000 ± 0.003 *p* = 0.958	0.007 ± 0.009 *p* = 0.466	−0.011 ± 0.043 *p* = 0.803	−0.024 ± 0.040 *p* = 0.549	0.003 ± 0.043 *p* = 0.939	−0.112 ± 0.172 *p* = 0.519
Transformed Handgrip Strength (kg)	0.904	40.523 *p* < 0.001	−0.030 ± 0.009 *p* = 0.001	1.901 ± 0.121 *p* < 0.001	0.004 ± 0.0.019 *p* = 0.834	0.002 ± 0.003 *p* = 0.680	−0.008 ± 0.008 *p* = 0.360	−0.018 ± 0.043 *p* = 0.683	−0.031 ± 0.042 *p* = 0.459	0.000 ± 0.032 *p* = 0.997	0.187 ± 0.197 *p* = 0.953

S.E. = standard error. ABPI = animal-based protein intake. Age: years. Gender: Women = 0, Men = 1. BMI: kg/m^2^. Relative energy intake: kcal/kg/day. Total protein intake was split at the median of percent energy from protein within both men and women; below median = 0, above median = 1. Nutritional variables were assessed using three-day food diaries. Summed isokinetic peak torque was calculated by adding the peak torques recorded during the isokinetic strength test, 60° per second for knee extension–flexion and 30° per second for plantar-dorsiflexion. Summed isokinetic endurance was calculated by adding total work performed during a 21-repetition test at 180° per second for the knee extension–flexion and 60° per second for plantar-dorsiflexion. Total repetitions performed during the 30-s chair stand test and handgrip strength were transformed using the square root function. The height of the chair for the 30-s chair stand test was 43 cm.

## Data Availability

Data is available as a Appendix A.

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
