# Peer review of "Measures Derived from Panoramic Ultrasonography and Animal-Based Protein Intake Are Related to Muscular Performance in Middle-Aged Adults"

_jcm, 2021, doi:10.3390/jcm10050988_

Round 1

Reviewer 1 Report

Abstract: Distinction needs to be made between ultrasonography and panoramic US since most would consider ultrasonography to be a broad category in which panoramic assessments fall under. So what type of ultrasonography are you referring to that may not be able to capture full CSA? Traditional muscle thickness assessments?

Line 20: I would suggest removing the statement regarding 75% being traditional for RF assessments since 50% is a decently common location for measurement now. Thomaes, T., et al., Reliability and validity of the ultrasound technique to measure the rectus femoris muscle diameter in older CAD-patients. Bmc Medical Imaging, 2012. 12.

Stratton et al. 2020, and much of Hester et al.’s work utilized 50% for RF CSA assessment.

Line 26: Add a clarifying statement that they were 3-day dietary food logs

Line 30: The r values should be added into the parentheses since you are discussing the relation.

Line 31: The phrase panoramic US seems suitable to assess sarcopenia does not seem to have much backing it up here and is too weak of a statement. I would suggest removing it or replacing it with something to the effect of Panoramic US shows promise as a method for assessing sarcopenia.

Introduction:

Line 37: The opening line is very clunky and should be reworded.

Line 40: Define what you mean by physical performance.

Line 46: The argument about BIA does not seem to add that much support to your argument. I would suggest removing it.

Since you discuss sarcopenia so much a brief statement about what the condition actually entails and which guidelines you are using to diagnose sarcopenia would be beneficial.

Line 58: I am not a fan of this argument about muscle size and ease of US capture. I would suggest removing it.

Line 64: There is a very good argument for investigating this relationship but with how prevalent panoramic measurements are currently I do not believe the discussion regarding transverse measurements is warranted.

Line 72: Remove “ we felt”

Line 91: Remove the parenthesize after the citation.

Line 94: Define what measures of muscular performance.

Methods:

Line 97: add from in between lab and October

Line 99: Remove “there were 2 sessions” also the sentence describing the variables starting on this line needs to be reworked. And is used to often.

Line 102: was this an additional 3-day food diary? If so this should be stated.

Line 107: Why did you use the 2011 PAR-Q when the newer versions are free online? Also what “more detailed health history questionnaire” are you referring to? The PAR-Q or something else? If it is the PAR-Q remove the statement I have here in quotations. If it is another questionnaire state which questionnaire that is.

Line 124: Did you determine leg dominance as well?

Line 126: Do you have reliability assessments between the 3 RAs that completed these assessments? If so that should be reported. Also, justification needs to be made for why US assessments were performed standing when this is far from the norm.

Discussion:

Line 472: You should state these positions as landmarks not traditional versus panoramic. So 50% versus 75%. Once again, be specific regarding which measures of muscular performance.

Line 483: remove “a limitation” also, limitations should be moved to the later portion of the discussion and all compiled together.

Line 519: please report again what their reliability was between sonographers.

Line 538: it would be more appropriate to assess protein intake as grams /kg than percentage of total energy intake. If protein intake is sufficient muscle mass will be better maintained even in a hypocaloric state. Hence why all current recommendations are given in g/kg.

doi: 10.1016/j.jamda.2013.05.021

Line 553: The fact that dietary intake was self reported should be included. It has been well established there is a large inherent error in these methods.

Author Response

Dear Reviewers and Editor:

Thank you for your careful notes and suggestions from improvement of our manuscript. We respectively offer our edits, rationale, or explanation for decision regarding methods and context of the study design, results and discussion. Our comments are in red below the reviewer comments listed here, starting with Reviewer number 1. Also, for revisions line numbers are noted. For ease of follow-up, we chose to use track changes in the Journal version of the manuscript.

Reviewer number 1

Abstract: Distinction needs to be made between ultrasonography and panoramic US since most would consider ultrasonography to be a broad category in which panoramic assessments fall under. So what type of ultrasonography are you referring to that may not be able to capture full CSA? Traditional muscle thickness assessments?

Thank you for this comment. We agree that panoramic assessments fall under “ultrasonography”. We originally attempted to use the term “traditional” to describe the use of a single standardized B-mode image. A novel comparison we made was with the use of the extension pack for panoramic image mode. We did not intend to use the term “traditional” as measurement “site” along the length of the upper leg (50% versus 75%). This may be where some confusion is and we will correct in the next submission. The rectus femoris (RF) is one of the few muscles that can easily be obtain in most individuals with a single B-mode ultrasound along the length of the anterior thigh. But even in some individuals with large muscles the width of the muscle is outside of view and if you attempt to keep all images views (depth/field of view) standardized it is problematic. However, other muscles (e.g., vastus lateralis, vastus intermedius) can be too large and outside of the  field of view. Thus, the ability to use the panoramic feature at specific sites to move across the muscle is advantageous. We first described this potential in Scott, Martin, Ploutz-Snyder, Matz, Caine, Downs, Hackney, Buxton, Ryder, and Ploutz-Snyder L (2017), J Cachexia, Sarcopenia, and Muscle. However, these measures were not associated with muscle strength or performance, which helps make the current submission novel.

Line 20: I would suggest removing the statement regarding 75% being traditional for RF assessments since 50% is a decently common location for measurement now. Thomaes, T., et al., Reliability and validity of the ultrasound technique to measure the rectus femoris muscle diameter in older CAD-patients. Bmc Medical Imaging, 2012. 12.

We agree with your comment. We have removed this from the abstract. Again, we did not mean for 75% to be labeled as “traditional”. We meant “traditional” to be used to described B-mode ultrasound. But there may be differences in the ability capture the RF muscle in one image at 50% in older CAD patients (likely with muscle loss) versus healthy younger, middle age, and older adults described in the current paper.

Stratton et al. 2020, and much of Hester et al.’s work utilized 50% for RF CSA assessment.

Our study ultrasound methods were established in 2016 prior to more recent recommendations and publications. Nevertheless, previous studies have demonstrated high test–retest reliability of ultrasound measures of muscle thickness of healthy adults taken in the standing position (Koskelo et al., 1991; Abe et al., 1994; Reimers et al., 1998). Recent studies also suggest the intraclass correlation coefficient (ICC) for standing measures of the anterior thigh muscles is 0.89 and recumbent is 0.90 (Thoirs et al. 2009).We are aware of the major concern is the fluid shift from standing to sitting, which has a more pronounced in the calf compared to the thigh (Berg et al., 1993). If we had pre-post type muscle ultrasound assessments we would have incorporated a 30+ fluid shift equilibration period, however, given this was a cross-sectional design and all participants were analyzed via ultrasound in a standing position we feel it is a valid comparison for thigh muscles (rectus femoris) as our methodology was consistent.

Line 26: Add a clarifying statement that they were 3-day dietary food logs

Yes agree that the addition of 3-day to line 26 would add clarification.

Line 30: The r values should be added into the parentheses since you are discussing the relation.

We agree with your notion, but think betas are better to present due to our mixed models. 

Line 31: The phrase panoramic US seems suitable to assess sarcopenia does not seem to have much backing it up here and is too weak of a statement. I would suggest removing it or replacing it with something to the effect of Panoramic US shows promise as a method for assessing sarcopenia.

Thank you for the suggestion.  “Panoramic US shows promise as a method for assessing sarcopenia” was used to replace line 31. 

Introduction:

Line 37: The opening line is very clunky and should be reworded.

Line 38-39 have been carefully rewritten to introduce the topic.

Line 40: Define what you mean by physical performance.

In Line 40-41 we list an example of 30-second chair stand.

Line 46: The argument about BIA does not seem to add that much support to your argument. I would suggest removing it.

Lines 46-48 regarding BIA were removed. BIA was removed from the entire paragraph.

Since you discuss sarcopenia so much a brief statement about what the condition actually entails and which guidelines you are using to diagnose sarcopenia would be beneficial.

The following was added to the “Introduction:”

According to the European Working Group on Sarcopenia in Older People 2, low muscle strength is the first criteria of sarcopenia, and low muscle mass or quality is the second; both must be assessed to be determined sarcopenia [2]. Low physical performance in addition to low muscle strength and quantity is considered severe sarcopenia [2].

Line 58: I am not a fan of this argument about muscle size and ease of US capture. I would suggest removing it.

The authors thank you for this comment. We were referring to the cost of measuring muscle size using MRI or CT ($500 per hour scan time) and the CT also introduces ionizing radiation. Dual energy x-ray absorptiometry also introduces a lower level of ionizing radiation but does not quantify muscle size as it only provides lean mass estimates (kg). Ultrasound provides a tool to measure muscle size without these barriers but we agree it is not easy as it is technologically challenging and requires significant practice. We will remove this sentence.

Line 64: There is a very good argument for investigating this relationship but with how prevalent panoramic measurements are currently I do not believe the discussion regarding transverse measurements is warranted.

We agree with this statement and will remove it from the manuscript.

Line 72: Remove “ we felt”

Agree.  Thanks for the suggestion.

Line 91: Remove the parenthesize after the citation.

This has been repaired.

Line 94: Define what measures of muscular performance.

Thank you for this comment. We have added to sentence:

Due to the differential impact that animal-based protein has on muscle protein syn-thesis, we secondarily sought to determine the relationship between animal-based protein intake (ABPI) and lower-body strength and endurance, handgrip strength, and 30-second chair stand performance, measures of muscular performance.

Methods:

Line 97: add from in between lab and October

This was completed

Line 99: Remove “there were 2 sessions” also the sentence describing the variables starting on this line needs to be reworked. And is used to often.

This was removed/reworked.

Line 102: was this an additional 3-day food diary? If so this should be stated.

Added “their” to line 103 to clarify that were was only ONE 3-day food diary.  Thanks for the suggestion.

Line 107: Why did you use the 2011 PAR-Q when the newer versions are free online? Also what “more detailed health history questionnaire” are you referring to? The PAR-Q or something else? If it is the PAR-Q remove the statement I have here in quotations. If it is another questionnaire state which questionnaire that is.

The PAR-Q 2011 was part of our original grant proposal and IRB approval and was used only for screening. This was prior to the PARQ+ which is now readily available and used for more recent students. The health history questionnaire was a study-specific, more comprehensive form used to collect information regarding conditions and chronic disease that might reveal subject characteristics that would make them ineligible for the study (e.g. condition that undermine muscle health; inability to use right leg, medications that might interfere metabolically with muscle health, etc.).

Line 124: Did you determine leg dominance as well?

All subjects were tested on right leg only, for consistency. We did not determine leg dominance.

Line 126: Do you have reliability assessments between the 3 RAs that completed these assessments? If so that should be reported. Also, justification needs to be made for why US assessments were performed standing when this is far from the norm.

All three research assistants completed reliability training prior to being allowed to be an operator for the testing in the study. The overall test-retest reliability of three images obtained by the research assistants using intraclass correlation coefficient  (ICCs) and 95% confidence interval is as follows:

Panoramic muscle thickness = 0.98 (0.90-0.95), B-mode muscle thickness = 0.98 (0.97-.99), Panoramic muscle area = 0.95 (0.93-0.96), B-mode muscle area =0.97 (0.97-0.98), Panoramic muscle echo intensity = 0.98 (.97-0.98), B-mode echo intensity = 0.81 (0.75-0.87).

Our study ultrasound methods were established in 2006 prior to more recent recommendations. Nevertheless, previous studies have demonstrated high test–retest reliability of ultrasound measures of muscle thickness of healthy adults taken in the standing position (Koskelo et al., 1991; Abe et al., 1994; Reimers et al., 1998). Recent studies also suggest the infraclass correlation for standing measures of the anterior thigh muscles is 0.89 and recumbent is 0.90 (Thoirs et al. 2009). Thus, there is a marginal difference. The major concern is the fluid shift from standing to sitting which has a more pronounced in the calf compared to the thigh (Berg et al., 1993). If we had pre-post type assessments we would have incorporated a 30+ fluid shift equilibration period, however, given this was a cross-sectional design and all participants were analyzed via ultrasound in a standing position it is a valid comparison for thigh muscles (Rectus femoris).

Discussion:

Line 472: You should state these positions as landmarks not traditional versus panoramic. So 50% versus 75%. Once again, be specific regarding which measures of muscular performance.

Thank you for this comment. We originally attempted to use the term “traditional” to describe the use of a single standardized B-mode image. A novel comparison we made was with the use of the extension pack for panoramic image mode. We did not intend to use the term “traditional” as measurement “site” along the length of the upper leg (50% versus 75%). This may be where some confusion is and we will correct in the next submission.

Line 483: remove “a limitation” also, limitations should be moved to the later portion of the discussion and all compiled together.

We moved to line 558-560 under limitations and compiled all limitations together near the end of the discussion.

Line 519: please report again what their reliability was between sonographers.

Panoramic muscle thickness = 0.98 (0.90-0.95), B-mode muscle thickness = 0.98 (0.97-.99), Panoramic muscle area = 0.95 (0.93-0.96), B-mode muscle area =0.97 (0.97-0.98), Panoramic muscle echo intensity = 0.98 (.97-0.98), B-mode echo intensity = 0.81 (0.75-0.87).

Line 538: it would be more appropriate to assess protein intake as grams /kg than percentage of total energy intake. If protein intake is sufficient muscle mass will be better maintained even in a hypocaloric state. Hence why all current recommendations are given in g/kg.

doi: 10.1016/j.jamda.2013.05.021

Yes, we agree. Frankly, in retrospect we find the line to be misleading as well. It has been replaced by:

Although recommendations for protein intake are made on a g/kg basis [37], one of advantages expressing intakes as percentages of energy intake is that one can control for relative energy intakes and for macronutrient composition in the same statistical model. There is a high degree of collinearity between relative intakes of macronutrients and relative energy intake. In fact, one of the main findings from Højfeldt and col-leagues study of older Danish adults was that relative protein intakes and relative energy intakes are related [17]. Collinearity can bias estimates of betas in multivariate analyses [38]. Although there is still a degree of collinearity between macronutrient intakes as percentages of energy and relative energy intakes, we addressed this issue by using the HC3 method of calculating standard errors which is more robust to colline-arity and heteroscedasticity [30].   

In addition, because we discuss collinearity in this section, we decided to add a table, now “Table 7,” which shows Pearson Product-Moment Correlations of macronutrient intakes, including animal-based protein, and relative energy intake.     

Reviewer 2 Report

 Dear Authors

 This study is to investigate that the panoramic image by ultrasonography may be a suitable method to measure skeletal muscle size and overall muscle mass, and seems suitable to assess sarcopenia. Moreover, the study shows that ABPI is related to better muscular performance.

However, I think this study has several questions and revisions as follows:

Minor points

  1. In table 5 and 6, ABP is written and should be revised into ABPI.
  2. The parts of Fig 2 are not shown and I cannot see the graph.
  3. In reference, articles of No. 21 to No. 35 are not shown and should be written.
  4. In table 1, 2, 5 and 6, because of using the Brown-Forsythe method in analyzes, and all values should be shown by median.

Major points

  1. Differences between the panoramic and traditional images and both of methods to measure are unclear in the section of 2.2 ultrasonography, page 3 and fig 1, page 4. If possible, both of panoramic and traditional images at 50% of leg length should be shown, and the panoramic feature in sentences, page 3 should be more explained.
  2. The results, ultrasonography and ABPI are related to muscle performances, are interesting. However, ABPI and age, gender, BMI, and so on, including ultrasonography should be analyzed in multiple regression analysis to investigate the relationship between ultrasonography, ABPI and muscle performances, like summed isokinetic peak torque, work, 30 second chair test and handgrip strength.
  3. The relationships between each of nutritional data from 3-day diaries and muscle performances and physical activities have been analyzed. However, the relationship between the nutritional data and ultrasonography should be analyzed.

Author Response

Dear Reviewers and Editor:

Thank you for your careful notes and suggestions from improvement of our manuscript. We respectively offer our edits, rationale, or explanation for decision regarding methods and context of the study design, results and discussion. Our comments are in red below the reviewer comments listed here, starting with Reviewer number 1. Also, for revisions line numbers are noted. For ease of follow-up, we chose to use track changes in the Journal version of the manuscript.

Reviewer number 2

Minor points

  1. In table 5 and 6, ABP is written and should be revised into ABPI.

Thank you.  This was changed.

  1. The parts of Fig 2 are not shown and I cannot see the graph.

We are very sorry about this. Figure 2 was made in landscape mode. We did not know it would format incorrectly. We made a new Figure 2 which should be amenable to the format/template.

  1. In reference, articles of No. 21 to No. 35 are not shown and should be written.

Again, we are very sorry. We are not sure why these were left out. They have been added.

  1. In table 1, 2, 5 and 6, because of using the Brown-Forsythe method in analyzes, and all values should be shown by median.

Yes, thanks. Medians have been added to these tables.

Major points

  1. Differences between the panoramic and traditional images and both of methods to measure are unclear in the section of 2.2 ultrasonography, page 3 and fig 1, page 4. If possible, both of panoramic and traditional images at 50% of leg length should be shown, and the panoramic feature in sentences, page 3 should be more explained.

We think some of this confusion is explained by a comment offered by reviewer #1, “traditional ultrasound” is a misnomer in our work. We took panoramic images at 50% of leg length, the typical place at which the rectus femoris is imaged with nonpanoramic ultrasonography. More specifically, as we indicated in our introduction, we choose to use panoramic ultrasonography at the 50% site because in those with greater muscle mass it is not always possible to image the entire transverse rectus femoris with nonpanoramic images. Although our methodology was established before the paper was published, this point was noted by Perkisas and colleagues’ (2018) article, “Application of ultrasound for muscle assessment in sarcopenia: towards standardized measurements.”   

In other words, we performed panoramic ultrasonography at 50% of leg length, and nonpanoramic B-mode ultrasonography at 75%. We did not take nonpanoramic images at 50% of leg length.

In effort to make these points clearer, we added the following to the section:

Images of the right rectus femoris muscle were captured using Philips ultrasound system (model HD11 XE; Bothell, WA) with a L12-5 50 mm linear array probe by three trained research assistants. Images were taken while participants were standing at marked sites 50% and 75% of the measured distance from the superior iliac spine of the hip to the lateral condyle of the knee. Participants were instructed to use their left leg as a base of support, while relaxing their right, resulting in a slight bend in the right knee. Previous works have shown high test–retest reliability of ultrasound measures of muscle thickness of healthy adults taken in the standing position [21,22]. A more recent study found the intraclass correlation coefficient (ICC) for standing measures of the anterior thigh muscles was 0.89, while the ICC for the same measures taken while participants were recumbent was 0.90 [23].Following generous application of ultrasonic gel, the probe was placed on the skin perpendicular to the leg and light, consistent pressure was applied to avoid excessive depression of the dermal surface until a full, clear image was obtained. The probe was removed from participants’ skin between each image acquisition, and markings were used to ensure the same area was measured. Because our participants were younger and likely have greater muscle size, the panoramic feature was used at the 50% site to record the entire transverse rectus femoris [11]. For panoramic ultrasonography, the lateral side of the right rectus femoris was identified, and the probe was moved medially until the entire transverse rectus femoris was recorded. B-mode image captures were taken at the 75% site where transverse sections of the rectus femoris are smaller. Three images were captured at each site using a frequency of 37 hz with a standardized depth of 7 cm and gain of 100%.

  1. The results, ultrasonography and ABPI are related to muscle performances, are interesting. However, ABPI and age, gender, BMI, and so on, including ultrasonography should be analyzed in multiple regression analysis to investigate the relationship between ultrasonography, ABPI and muscle performances, like summed isokinetic peak torque, work, 30 second chair test and handgrip strength.

Although we find the idea of evaluating a regression model including both measures from ultrasonography and those from our dietary analyses interesting, we think it is beyond the scope of our research questions. We sought to evaluate the relationship between measures from different methods of ultrasonography and muscular performance. We also sought to determine the association between animal-based protein intake and measures from ultrasonography and muscular performance. We did not hypothesize about using both measures from ultrasound and animal-based protein intake, or another dietary variable, to predict muscular performance.       

  1. The relationships between each of nutritional data from 3-day diaries and muscle performances and physical activities have been analyzed. However, the relationship between the nutritional data and ultrasonography should be analyzed.

Yes, before data collection, it was our goal to perform these analyses as well. However, three male participants do not have measures from ultrasonography because the ultrasound machine failed. All three of these men were, coincidently, above the median for animal-based protein intake as a percentage of energy. Due to this effect on our cell sizes in our mixed models, reducing men above the median to 17, we decided not to perform these analyses. We added the following to section 2.6.3

For our descriptive statistics, we described the four groups from the secondary analyses in our all of our tables, even though we choose not to investigate the association between ABPI and measures from ultrasonography because the three of men who were precluded from ultrasonography were, coincidently, above the median for animal-based protein intake as a percentage of energy. Within these tables, we chose to use the Brown-Forsythe method for comparisons, because we did not assume equal variances. We compared those above the median of ABPI as a percentage of energy to those below the median within each gender, so we did not adjust for multiple comparisons

Sincerely,

The Authors

Round 2

Reviewer 1 Report

N/A

Author Response

We attached a letter

Reviewer 2 Report

Dear Authors

Thank you very much for your comments and revisions. I think that you should revise minor points.

  1. In table 1, 2, 5 and 6, you added both of median and average data to each table, but I think that median data only would be needed, so please revise them.

From reviewer.

Author Response

We attached a letter stating the following:

These tables have been revised. Means and S.E.M. have been removed
